# Interaction of Epigallocatechin Gallate and Quercetin with Spike Glycoprotein (S-Glycoprotein) of SARS-CoV-2: In Silico Study

**DOI:** 10.3390/biomedicines10123074

**Published:** 2022-11-29

**Authors:** Mehran Alavi, M. R. Mozafari, Saba Ghaemi, Morahem Ashengroph, Fatemeh Hasanzadeh Davarani, Mohammadreza Mohammadabadi

**Affiliations:** 1Department of Biological Science, Faculty of Science, University of Kurdistan, Kurdistan 6617715175, Iran; 2Nanobiotechnology Department, Faculty of Innovative Science and Technology, Razi University, Kermanshah 6714414971, Iran; 3Australasian Nanoscience and Nanotechnology Initiative (ANNI), Monash University LPO, Clayton, VIC 3168, Australia; 4Research Committee of Medical School, Alborz University of Medical Science, Karaj 3149779453, Iran; 5Department of Plant Pathology, Rafsanjan Branch, Islamic Azad University, Rafsanjan 7718897111, Iran; 6Department of Animal Science, Faculty of Agriculture, Shahid Bahonar University of Kerman, Kerman 7616913439, Iran

**Keywords:** SARS-CoV-2, transmembrane spike glycoprotein, severe pneumonia, natural compounds, antiviral activity

## Abstract

Severe acute respiratory syndrome (SARS)-CoV-2 from the family Coronaviridae is the cause of the outbreak of severe pneumonia, known as coronavirus disease 2019 (COVID-19), which was first recognized in 2019. Various potential antiviral drugs have been presented to hinder SARS-CoV-2 or treat COVID-19 disease. Side effects of these drugs are among the main complicated issues for patients. Natural compounds, specifically primary and secondary herbal metabolites, may be considered as alternative options to provide therapeutic activity and reduce cytotoxicity. Phenolic materials such as epigallocatechin gallate (EGCG, polyphenol) and quercetin have shown antibacterial, antifungal, antiviral, anticancer, and anti-inflammatory effects in vitro and in vivo. Therefore, in this study, molecular docking was applied to measure the docking property of epigallocatechin gallate and quercetin towards the transmembrane spike (S) glycoprotein of SARS-CoV-2. Results of the present study showed Vina scores of −9.9 and −8.3 obtained for EGCG and quercetin by CB-Dock. In the case of EGCG, four hydrogen bonds of OG1, OD2, O3, and O13 atoms interacted with the Threonine (THR778) and Aspartic acid (ASP867) amino acids of the spike glycoprotein (6VSB). According to these results, epigallocatechin gallate and quercetin can be considered potent therapeutic compounds for addressing viral diseases.

## 1. Introduction

Fighting against viral, bacterial, and parasitic infections has been an ongoing medical challenge of modern times [1,2,3,4]. A major concern in this regard is emerging mutant strains having a wide range of resistance to conventional drugs [5,6,7,8]. On the other hand, the treatment of cancer and metabolic diseases (e.g., type 2 diabetes, heart disease, and stroke) has certain drawbacks, particularly adverse side effects associated with therapeutic strategies [9,10]. These drawbacks can be overcome by new effective therapies based on micro- and nanomaterials with lower or ideally no side effects [11,12,13,14,15,16,17,18,19]. In the case of viral infections, severe acute respiratory syndrome (SARS)-CoV-2, from the family Coronaviridae, is the cause of the outbreak of fatal pneumonia, coronavirus disease 2019 (COVID-19), which was first recognized in 2019 in Wuhan, China [20,21,22,23]. SARS-CoV-2 has a transmembrane spike (S) glycoprotein with two functional subunits: the S1 subunit having the receptor-binding domain (RBD), which can bind to the host cell receptor, and the S2 subunit with the ability to fuse with the host cell membranes (Figure 1a,b) [24,25].

The worldwide spread of SARS-CoV-2 has resulted in an urgent need to find effective targets for the hindering of COVID-19 and associated viruses. Accordingly, blocking of spike (S) glycoprotein is critical for the inactivation of SARS-CoV-2 before the initiation of a cytokine storm, the production of many inflammatory signals by the immune system, which can result in organ failure and death of patients [27]. The S glycoproteins and the SARS-CoV-2 endoribonuclease Nsp15 have also been shown to be excellent targets for the development of vaccines against coronaviruses [28,29,30]. Furthermore, several molecules and moieties have been identified as antiviral agents thus far. Some of these therapeutic compounds along with their mechanisms of action are listed in Table 1 [31,32,33,34,35,36,37]. However, there are significant adverse side effects associated with these molecules and therapeutic agents. As a result, it is a matter of urgency to find biocompatible therapeutic agents against novel viral strains, particularly SARS-CoV-2 and other coronaviruses.

The natural compound–virus interface and the corresponding viral responses are crucial for determining the level of antiviral activity for each natural therapeutic agent [51,52]. Natural compounds, particularly secondary metabolites of medicinal plants associated with phenolic compounds having benzene rings with one or more hydroxyl substituents, terpenoids (derived from the five-carbon compound isoprene), saponins (triterpene glycosidic compounds), alkaloids (organic compounds containing at least one nitrogen atom), and glucosinolates (sulfur-containing metabolites), have demonstrated prominent antibacterial, antioxidant, anticancer, antiviral, antiarthritic, anti-Alzheimer’s, cardiovascular, and wound-healing activities [53,54]. Specifically, quercetin (C_15_H_10_O_7_), extracted mainly from green tea, grapes, apples, berries, and onions, is a flavonol related to the flavonoid group of phenolic compounds, which is known to have antimicrobial, anti-inflammatory, antioxidant, and anticancer activities; and apoptosis-inducing effect; and therapeutic effects on metabolic diseases such as nonalcoholic fatty liver disease, diabetes, and hyperlipidemia [55,56,57,58]. Effective doses for this metabolite have been reported as 500–1000 mg/day or 50 and 75 mg/kg [57]. In the case of epigallocatechin gallate (epigallocatechin-3-gallate (EGCG; C_22_H_18_O_11_) extracted especially from green tea (~103 mg/g) and black tea (24.7 mg/g), a variety of therapeutic effects have been reported that include antibacterial, antioxidant, anticancer, anti-inflammatory, antiobesity, antidiabetic, chemopreventive, and antiviral activities [59,60,61,62,63,64]. For example, inhibition of the uridylate-specific endoribonuclease Nsp15 from SARS-CoV-2 has been found for three bioactive compounds of EGCG, quercetin, and baicalin [61]. It should be noted that despite in silico studies, new techniques including tissue diffusion chambers, single-organ chips, and body-on-a-chip can help the clinical development of natural drugs [65].

For the elucidation of drug–target interaction and optimization of therapeutic outcome, comprehensive in vitro and in vivo investigations along with relevant computational studies are required. In silico study is one of the main methods to evaluate the activity of new drugs and bioactive agents by computational structure-based drug discovery [66]. According to the above discussion, docking of epigallocatechin gallate and quercetin towards spike (S) glycoprotein of SARS-CoV-2 was evaluated by three docking programs: CB-Dock, DockThor, EDock, and AutoDock Vina 1.5.7 (ADV).

## 2. Materials and Methods

Epigallocatechin-3-gallate (EGCG) and quercetin were selected as ligands, and spike glycoprotein (PDB ID: 6VSB) and Nsp15, a uridine-specific endoribonuclease (PDB ID: 6VWW), were selected as the receptors (Figure 2a–d). In order to minimize the final structures of epigallocatechin gallate and quercetin and remove all the water molecules, the UCSF Chimera1.12 program (a program for the interactive analysis and visualization of molecular structures including trajectories, sequence alignments, and density maps) was employed [67]. The web servers of Cavity-Detection Guided Blind Docking (CB-Dock) (http://clab.labshare.cn/cb-dock/php/) (accessed on 1 September 2022) [68], DockThor (https://www.dockthor.lncc.br/v2/) (accessed on 1 September 2022) [69], EDock (https://zhanggroup.org/EDock/) (accessed on 1 September 2022) [70], and ADV 1.5.7 [71] were applied to evaluate and compare molecular docking. The affinity of docked quercetin and EGCG with spike glycoprotein was presented as binding energy (kcal/mol). Results of docking interaction were visualized by BIOVIA discovery studio 2016 (San Diego, CA, USA) (Figure 2a,b). Gaussian 5.0.8 software was employed to optimize geometry and to determine the electric field potential, lowest unoccupied molecular orbital (LUMO), and the highest occupied molecular orbital (HOMO) of quercetin in the ground state, Hartee-fock at default spin and basis set of 3–21G [72].

## 3. Results and Discussion

According to the CB-DOCK results, for EGCG towards the receptor of 6VSB, cavity volumes were 5396, 8798, 11,401, 7201, and 2780 Å^3^ for Vina scores of −9.9, −9.1, −9, −8.9, and −8.4, respectively (Table 2). For this metabolite, higher score of −8.8 and cavity volume of 1021 Å^3^ were observed toward 6VWW (Table 3).

Four hydrogen bonds between OG1, OD2, O3, and O13 of EGCG with THR778 and ASP867 amino acids of the 6VSB were the main chemical interactions between EGCG and the receptor as depicted in Figure 3a,b. As illustrated in Figure 4a,b, the LYS71 (H-bond), LYS90, GLY165, VAL166 (H-bond), THR167 (H-bond), ARG199, ASN200, GLU203, ASP268, ILE270, PRO271, MET272 (H-bond), ASP273, SER274, LYS277 (H-bond), and TYR279 (H-bond) amino acids of 6VWW contributed in the interaction with EGCG. In the case of quercetin and the receptor of 6VSB, cavity sizes were 2780, 8798, 11,401, 5396, and 7201 Å^3^ for Vina scores of −8.3, −8.2, −8.1, −8.1, and −7.7, respectively (Table 4). This docking server revealed that amino acids TRP886, TYR904, GLY908, GLY1035, GLN1036, LYS1038, GLY908, ILE909, GLN1036, SER1037, LYS1038, VAL1040, GLY1046, TYR1047, and HIS1048 of the spike glycoprotein had docking interaction with the quercetin metabolite (Figure 5a,b). Additionally, amino acids GLU69, LYS71 (H-bond), LYS90 (H-bond), THR196 (H-bond), SER198 (H-bond), ARG199, ASN200 (H-bond), LEU252 (H-bond), ASP273, SER274, THR275, LYS277, VAL295, ILE296 (H-bond), and ASP297 of 6VWW interacted with quercetin with a higher score of −8.2 (Figure 6a,b and Table 5).

According to the results of DockThor docking for 6VSB, affinity, total energy, van der Waals (vdW) energy, and electronic energy for EGCG against the receptor were −7.287 kcal/mol, 14.876 kcal/mol, −2.202, and −44.070 eV, respectively. In the case of quercetin, affinity, total energy, vdW energy, and electronic energy were −7.468 kcal/mol, 10.141 kcal/mol, −11.502, and −25.191 eV, respectively. In the case of 6VWW, affinity, total energy, vdW energy, and electronic energy were −7.056 kcal/mol, 26.627 kcal/mol, −3.350, and −30.466 eV for the EGCG ligand, respectively. Moreover, the affinity, total energy, vdW energy, and electronic energy for quercetin towards 6VWW were −6.891 kcal/mol, 18.233 kcal/mol, −2.538, and −27.190 eV, respectively. In a comparative study, quercetin and quercetin pentaacetate were evaluated against the human respiratory syncytial virus (hRSV) F-protein by in silico analysis. In that study, researchers discovered that acetylation of quercetin improves anti-hRSV activity, as quercetin pentaacetate had a lower binding energy with better stability with the value of ΔG= −14.22 kcal/mol in hindering F-protein and thus reducing hRSV adhesion [73]. Based on the EDock results, for EGCG and quercetin, three amino acids, namely ARG812, LEU813, and LEU816, of the receptor showed interaction with the active site of the spike glycoprotein (Figure 7a,b). Figure 7c,d show predicted binding residues of 6VWW with EGCG and quercetin.

Van der Waals interaction and hydrogen bonding were indicated for this interaction. 3CLpro (3-chymotrypsin-like protease), the main protease with the critical role in cleaving pp1a and pp1ab polyproteins, can be selected as the main target for the inactivation of SARS-CoV-2. In this respect, 73 bioactive compounds related to the medicinal plant Withania spp. were screened against 3CLpro. A study by Verma and co-workers revealed that there was more negative energy for withacoagulin H (−63.463 KJ/mol) than for other natural compounds [74]. Molecular docking of three secondary metabolites extracted from the n-butanol and ethyl acetate fractions of Amphilophium paniculatum from the Bignoniaceae family toward the SARS-CoV-2 main protease (Mpro) was evaluated. According to the results, eight molecules, namely luteolin, luteolin 7-O-β-glucopyranoside (cynaroside), acacetin 7-O-β-rutinoside (linarin), acteoside (verbascoside), Isoacteoside (Isoverbascoside), (+)-Lyoniresinol 3α-O-β-glucopyranoside, (−)-Lyoniresinol 3α-O-β-glucopyranoside, and amphipaniculoside A, were found with lower binding energies of −8.34, −9.54, −8.54, −8.33, −8.46, −7.95, −7.45, and −7.56 kcal/mol, respectively. The major bond types for luteolin 7-O-β-glucopyranoside were hydrogen bonds (GLU166, CYS145, GLY143, ASN142, ASN142, ASN142) and π–π interactions (HIS41 and HIS41) [75]. In a similar study, the docking of ten compounds (9-dihydroxyl-2-O-(z)-cinnamoyl-7-methoxy-Aloesin, aloe-emodin, aloin A, aloin B, elgonica dimer A, feralolide, isoAloeresin, aloeresin, 7-O-methylAloeresin, and chrysophanol) related to the *Aloe vera* plant species was evaluated toward 3CLpro. Three bioactive agents, namely feralolide, aloeresin, and 9-dihydroxyl-2-O-(z)-cinnamoyl-7-methoxy-Aloesin, exhibited higher affinity for 3CLpro with binding energies of −7.9, −7.7, and −7.7 kcal/mol, respectively, compared to standard drugs of lopinavir (−8.4 kcal/mol) and nelfinavir (−8.1 kcal/mol) [76]. Moreover, according to the results of a comprehensive docking study, coagulins, withanolides, pseudojervine, and kamalachalcone groups of triterpenoid compounds demonstrated the potential ability to block surface amino acids of the spike protein of SARS-CoV-2 (the head of S1 which binds to the cellular receptor hACE2) [77]. In another study, a flavonoid (i.e., rutin) showed inhibition of major proteins of SARS-CoV-2, namely the spike (S)-protein (S1 subunit of S-protein), papain-like protease (PLpro), main protease (Mpro), and RNA-dependent RNA polymerase (RdRp), with binding energies of −7.9, −7.7, −8.9, and −8.6 kcal/mol, respectively. The numbers of hydrogen bonds were 3, 9, 10, and 6 for Mpro, RdRp, PLpro, and S1 subunit of S-protein, respectively [78].

Al-Karmalawy and coworkers (2021) [79] employed molecular docking to investigate the affinity of 14 angiotensin-converting enzyme inhibitors (ACEIs) towards the SARS-CoV-2 binding site of chimeric receptor-binding domain bound by its receptor human angiotensin-converting enzyme 2 (hACE2). For this study, alacepril, captopril, zofenopril, enalapril, ramipril, quinapril, perindopril, lisinopril, benazepril, imidapril, trandolapril, cilazapril, fosinopril, and moexipril were the tested ligands, and N-Acetyl-D-Glucosamine (NAG) was employed as a reference ligand. This study revealed that there were the same binding modes for lisinopril, alacepril, and NAG. Additionally, the binding scores for lisinopril and alacepril were −4.7 and −5.1 with two hydrogen bonds, respectively [79]. In another similar study, in which lopinavir (a protease inhibitor drug) was used as a reference drug (with a MolDock score of −114.628), the antiparasitic drug ivermectin exhibited a MolDock score of −114.860, and the formation of three hydrogen bonds with Asn2033, Asn151, and Asp153 amino acid residues was detected [80]. There was a MolDock score of −95.414 for hydroxychloroquine with interactions of three hydrogen bonds with Asn203, Gln109, and Ser158 amino acid residues. Moreover, chloroquine exhibited a MolDock score of −93.634 and two hydrogen bonds with Ser158 [80]. Molecular docking of three natural compounds, namely chrysin (flavonoid), hesperidin (flavonoid), and emodin (anthraquinone), against the ACE2 protein and the complexed structure of the ACE2 protein and spike protein was investigated in a comparative study. The binding energies for hesperidin, chrysin, and emodin were −8.99, −6.87, and −6.19 kcal/mol toward the bound spike protein and ACE2 receptor, respectively. Depending on the results, the binding sites of ACE2 protein for hesperidin and spike protein were in different sites of the ACE2 protein, and this metabolite can lead to instability of the bound structure of spike protein and ACE2 by modulating the binding energy of the bound structure of the spike protein and ACE2. In addition, hesperidin binds at the LYS74, ALA71, SER44, and ASN63 amino acids of ACE2 with stabilized docking by two hydrogen bonds at PHE457 of the spike protein with a distance of 2.618 Å and GLU455 of spike protein with a bond length of 2.067 Å [81]. Based on the results of ADV (Table 6), higher binding affinities towards 6VSB and 6VWW were found for EGCG, namely −9.9 and −7.3 kcal/mol, compared to those found for quercetin with the values of −7.6 and −6.1 kcal/mol, respectively. In a comparative study, gallocatechin gallate, EGCG, quercetin, puerarin, and daidzein flavonoids exhibited IC_50_ (50% inhibitory concentration or half-maximal effective concentration) values of 47, 73, 73, 381, and 351 µM, respectively, for inhibition of SARS-CoV replication. Furthermore, docking scores of −14.1, −11.7, −10.2, −11.3, and −8.6 have been found for gallocatechin gallate, EGCG, quercetin, puerarin, and daidzein, respectively [82]. In another study, EGCG had an IC_50_ value of 0.874 µM with the binding energy of −7.9 kcal/mol against 3CL^pro^ SARS-CoV-2 [83]. In the case of EGCG, plaque reduction neutralization antibody tests confirmed the inhibition of a SARS-CoV-2 strain at PRNT_50_ = 0.20 μM titer [61].

### Molecular Electrostatic Potential

Based on the optimized geometry obtained using GaussView 5.0.8 software, the electric field potentials of EGCG and quercetin were identified as electrophilic and nucleophilic regions by the ground state method, with Hartee-fock at default spin and basis set of 3–21G. As depicted in Figure 8a,b, EGCG is a polyphenol, the ester of epigallocatechin and gallic acid, and is composed of a 22-carbon skeleton bonded by 18 hydrogens and 11 oxygens.

Quercetin as a flavonoid compound has three aromatic rings with a 15-carbon skeleton bonded by oxygen atoms encapsulated in a heterocyclic ring (Figure 9a,b) [60,61,62]. In Figure 8c,d and Figure 9c,d, a higher density of electrons is shown in red color and a lower density of electrons is shown in blue color. The aromatic ketone of EGCG and quercetin with more electrons can be attacked by electrophilic residues in ligand-binding cavities. In contrast, blue regions are suitable sites for nucleophilic attacks [84]. In this way, three amino acids with basic side chains and positive charge, namely lysine (a propylamine substituent on the β-carbon), arginine (guanidino group), and histidine (imidazole functional group) can contribute to this interaction [85,86]. The high kinetic stability of a compound can be a result of a large HOMO-LUMO gap [87].

## 4. Conclusions

The global spread of SARS-CoV-2 has led to an urgent requirement for finding effective targets for eradicating this virus. In silico study is one of the major strategies for surveying the activity of new drugs and bioactive compounds by computational structure-based drug discovery because it is cost-effective relative to the experimental studies. Active and passive targeting of viruses by new effective biocompatible materials is a vital measure for hindering viral infections, specifically SARS-CoV-2 infections. The natural compound–virus interface and the corresponding viral responses are crucial for determining the level of antiviral activity for each natural therapeutic agent. It is clear in the present time that there are no certain effective therapies for COVID-19, while the side effects of available antiviral drugs constitute a great disadvantage. In this in silico study, CB-Dock exhibited Vina scores of −9.9 and −8.8 for EGCG against 6VSB and 6VWW and −8.3 and −8.2 for quercetin against 6VSB and 6VWW. DockThor showed affinity values of −7.056 kcal/mol and −6.891 kcal/mol for EGCG and quercetin toward 6VWW. According to the result of ADV, higher binding affinities towards 6VSB and 6VWW were found for EGCG (−9.9 and −7.3 kcal/mol, respectively) than for quercetin (−7.6 and −6.1 kcal/mol, respectively). Additionally, molecular electrostatic potential showed that aromatic ketone of EGCG and quercetin with a higher density of electrons can be attacked by the electrophilic amino acids of the spike glycoprotein of SARS-CoV-2. It should be noted that docking comparison of EGCG and quercetin with other main secondary metabolites is indispensable. Overall, this study showed that EGCG had stronger affinities toward two receptors 6VSB and 6VWW compared to quercetin, which may be considered for formulation as micro- and nanosized antiviral drugs against SARS-CoV-2 infections. Based on the results of molecular electrostatic potential, the aromatic ketone of EGCG and quercetin with more electrons can be attacked by electrophilic residues in ligand-binding cavities of 6VWW and 6VSB. In this regard, three amino acids with basic side chains and positive charge, namely lysine, arginine, and histidine, can contribute to this interaction. There are main limitations including low tissue exposure/selectivity and low specificity/potency for clinical applications of natural drugs. Therefore, it is critical to obtain effective doses and their stability (half-life) in physiological conditions, which may be possible using nanoformulations (solid lipid nanoparticles, liposomes, and polymeric nanoparticles) of these natural compounds. Moreover, new technologies involving tissue diffusion chambers, single-organ chips, and body-on-a-chip can accelerate clinical development of natural drugs.

## Figures and Tables

**Figure 1 biomedicines-10-03074-f001:**
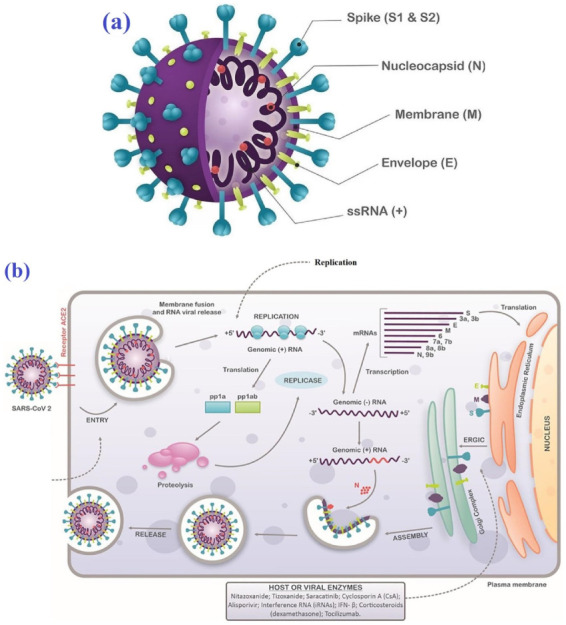
(**a**) Main parts of SARS-CoV-2, including spike (S1 and S2 subunits), nucleocapsid, membrane, envelope, and ssRNA. (**b**) Schematic image showing the replication cycle of SARS-CoV-2 in host cells. Distributed under the terms of the Creative Commons Attribution License (CC BY) [26].

**Figure 2 biomedicines-10-03074-f002:**
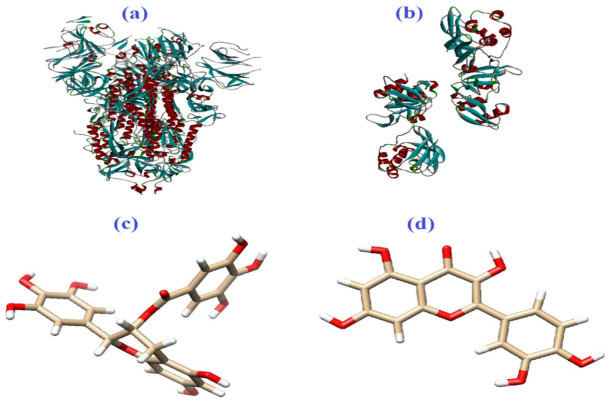
(**a**) Spike glycoprotein (PDB ID: 6VSB), (**b**) Nsp15 (PDB ID: 6VWW), (**c**) epigallocatechin-3-gallate (EGCG), and (**d**) quercetin.

**Figure 3 biomedicines-10-03074-f003:**
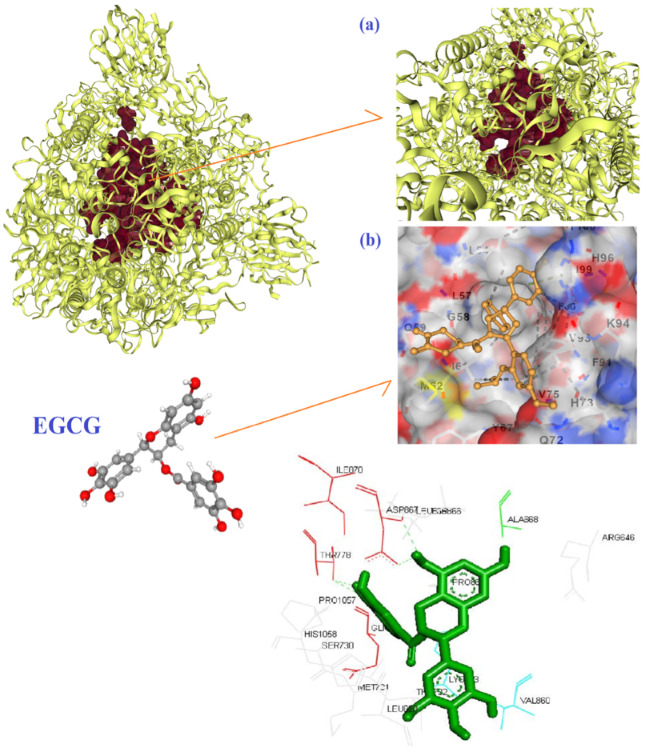
The best mode of the active site for the receptor of 6VSB (**a**) and its interaction with EGCG (**b**) based on the results of CB-Dock.

**Figure 4 biomedicines-10-03074-f004:**
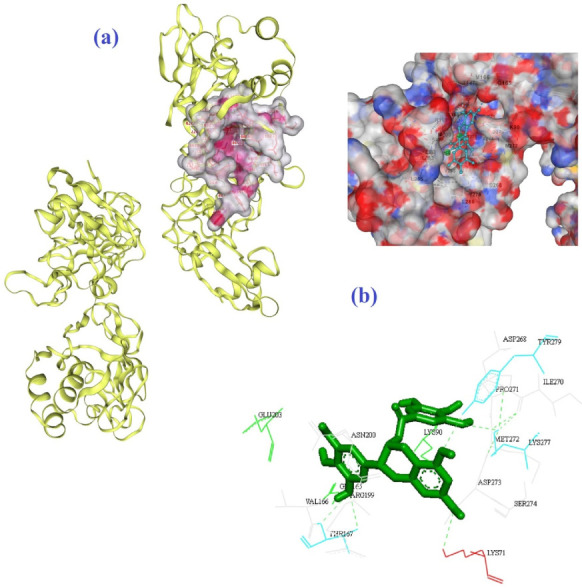
The best mode of the active site for the receptor of 6VWW (**a**) and its interaction with EGCG (**b**) based on the results of CB-Dock.

**Figure 5 biomedicines-10-03074-f005:**
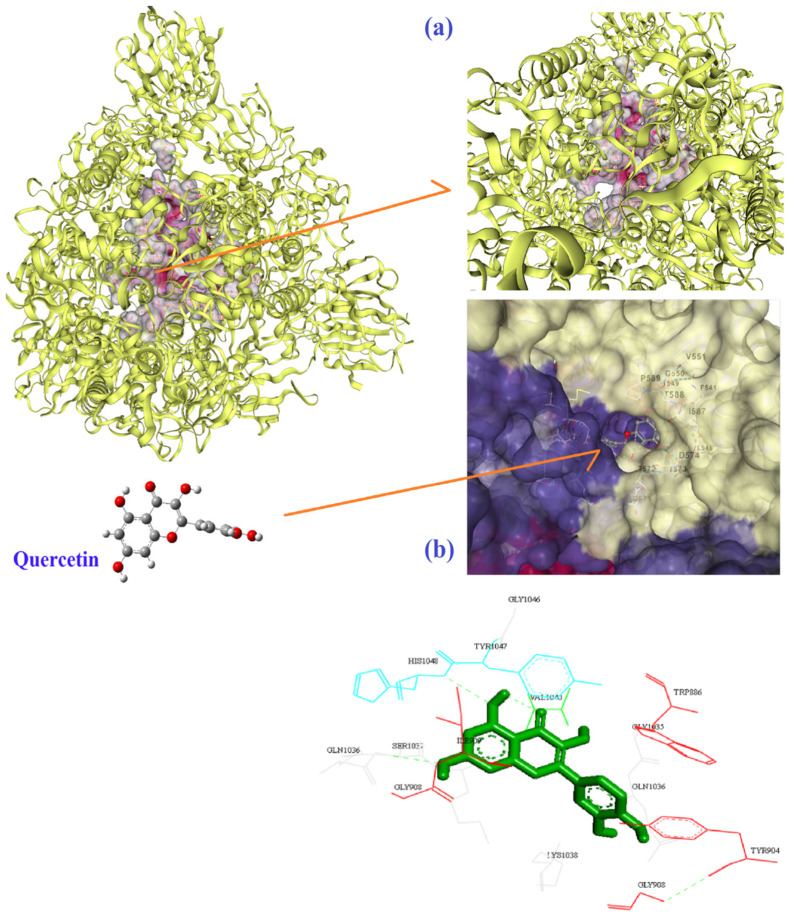
The best mode of the active site for the receptor of 6VSB (**a**) and its interaction with quercetin (**b**) according to the results of CB-Dock.

**Figure 6 biomedicines-10-03074-f006:**
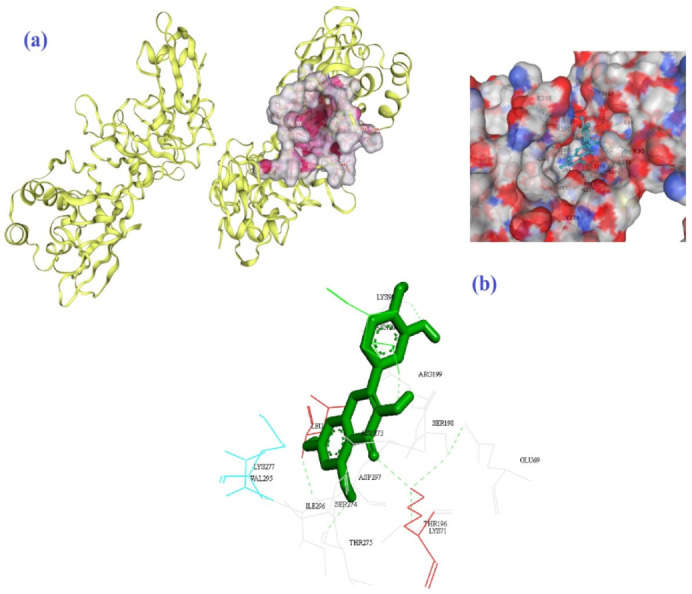
The best mode of the active site for the receptor of 6VWW (**a**) and its interaction with quercetin (**b**) according to the results of CB-Dock.

**Figure 7 biomedicines-10-03074-f007:**
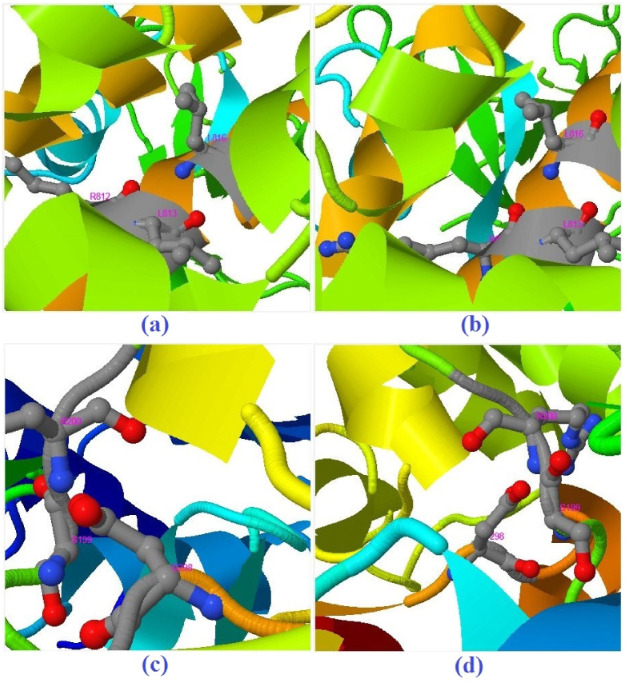
The best modes for the interaction between 6VSB and (**a**) EGCG and (**b**) quercetin as well as between 6VWW and (**c**) EGCG and (**d**) quercetin ligands based on the results of EDock.

**Figure 8 biomedicines-10-03074-f008:**
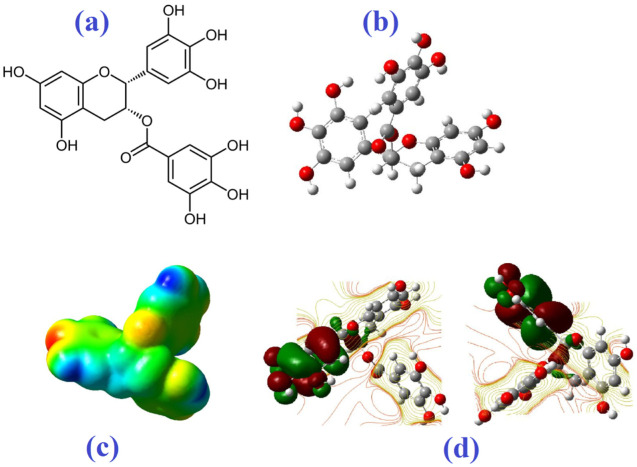
The molecular structure (**a**,**b**), total electron density maps (**c**), 3D model of HOMO-LUMO molecular orbitals, and (**d**) contour plots (side view: **left**; front view: **right**) of EGCG.

**Figure 9 biomedicines-10-03074-f009:**
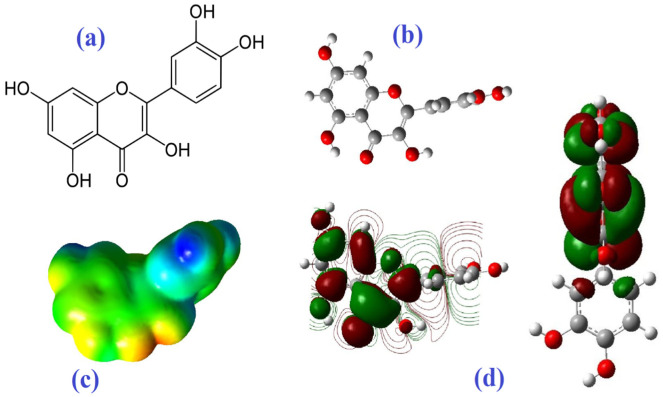
The molecular structure (**a**,**b**), total electron density maps (**c**), 3D model of HOMO-LUMO molecular orbitals, and (**d**) contour plots (side view: **left**; front view: **right**) of quercetin.

**Table 1 biomedicines-10-03074-t001:** Some of the identified molecules and compounds with antiviral activity.

Molecule	Mode of Action	Ref.
chloroquine/hydroxychloroquine	inhibiting glycosylation of host receptors	[38]
azithromycin	antibiotic/anti-inflammatory activities	[39]
lopinavir/ritonavir	inactivation of the viral 3CL protease	[40]
ribavirin	blocking RNA-dependent RNA polymerase	[41]
tocilizumab	inhibition of IL-6 signaling	[42]
baricitinib/remdesivir	blocking RNA-dependent RNA polymerase	[43]
favipiravir	selective inhibition of viral RNA polymerase	[44]
abidol	broad-spectrum antiviral compound	[45]
ruxolitinib	competitively blocking the ATP-binding catalytic site on Janus kinases 1 and 2	[46]
teicoplanin	an antibiotic applied in the prophylaxis and therapy of bacterial infections caused by Gram-positive bacteria	[47]
ivermectin	inhibition of the nuclear transport of viral proteins	[48]
corticosteroid	immunosuppressive and anti-inflammatory activities	[49]
doxycycline	blocking bacterial protein synthesis via binding to the 30S ribosomal subunit	[50]

**Table 2 biomedicines-10-03074-t002:** Binding mode and the related cavity size based on the results of CB-Dock for EGCG and the receptor of 6VSB.

Vina Score	Cavity Volume (Å^3^)	Center			Size		
		x	y	z	x	y	z
−9.9	5396	207	244	243	35	23	35
−9.1	8798	252	231	233	32	35	35
−9	11,401	227	228	172	35	33	35
−8.9	7201	224	221	215	31	35	34
−8.4	2780	225	249	213	29	23	35

**Table 3 biomedicines-10-03074-t003:** Binding mode and the related cavity size based on the results of CB-Dock for EGCG and the receptor of 6VWW.

Vina Score	Cavity Volume (Å^3^)	Center			Size		
		x	y	z	x	y	z
−8.8	1021	−68	28	25	23	23	23
−8.7	1074	−73	26	−30	23	23	23
−8.2	627	−56	24	20	23	23	23
−7.7	615	−81	18	−21	23	23	23
−7.6	666	−53	32	−4	23	23	23

**Table 4 biomedicines-10-03074-t004:** Binding mode and the related cavity size based on the results of CB-Dock for quercetin and the receptor of 6VSB.

Vina Score	Cavity Volume (Å^3^)	Center			Size		
		x	y	z	x	y	z
−8.3	2780	225	250	213	29	21	35
−8.2	8798	253	232	233	32	35	35
−8.1	11,401	227	229	172	35	33	35
−8.1	5396	207	245	243	35	28	35
−7.7	7201	225	222	215	31	35	34

**Table 5 biomedicines-10-03074-t005:** Binding mode and the related cavity size based on the results of CB-Dock for quercetin and the receptor of 6VWW.

Vina Score	Cavity Volume (Å^3^)	Center			Size		
		x	y	z	x	y	z
−8.2	1021	−74	26	−30	21	21	21
−7.7	631	−56	24	20	21	21	21
−7.3	605	−53	21	−13	21	21	21
−6.6	641	−82	18	−21	21	21	21
−6.1	667	−53	32	−4	21	21	21

**Table 6 biomedicines-10-03074-t006:** The docking results of ADV for EGCG and quercetin toward 6VSB and 6VWW. Only two best modes are presented for each compound (RMSD/L.B: root-mean-square deviation, lower bound; RMSD/U.B: root-mean-square deviation, upper bound).

Ligand	Binding Affinity (kcal/mol) for 6VSB	Binding Affinity (kcal/mol) for 6VWW
EGCG	−9.9	−7.3
	−9.8	−7.2
Quercetin	−7.6	−6.1
	−7.4	−5.9

## Data Availability

Not applicable.

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
