# Peer review of "Interaction of Epigallocatechin Gallate and Quercetin with Spike Glycoprotein (S-Glycoprotein) of SARS-CoV-2: In Silico Study"

_biomedicines, 2022, doi:10.3390/biomedicines10123074_

Round 1

Reviewer 1 Report

1-    Reduce the number of keywords to 5 keywords. In the current state, there are too many.

2-    Figure 1 has a low resolution. Increase the resolution of the figure or replace it with a higher-quality image.

3-    The introduction is well-written, but there must be explained in some parts that the following references can be used: “Green synthesis of Mg0. 99 Zn0. 01O nanoparticles for the fabrication of κ-Carrageenan/NaCMC hydrogel in order to deliver catechin. Polymers12(4), 861”. And also “Effect of zinc content on structural, functional, morphological, and thermal properties of kappa-carrageenan/NaCMC nanocomposites. Polymer Testing93, 106922.”

4-    Table 1 needs relevant references for each line.

5-    Clarify the novelty of the research.

6-    Line 99, what is UCSF? Write its complete form for the first time using.

7-     Explain about UCSF Chimera1.12 program, and in the Materials section, there is no need to use references or links.

8-    The resolution of all figures is too low. Draw the structures again and present them in high resolution.

9-    There are some old references that can be replaced with newly published papers.

Author Response

Dear Reviewer;
Thanks for your instructive suggestions / comments. Please find our responses to your comments / suggestions below (changes are highlighted in the manuscript in Green colour):

1-    Reduce the number of keywords to 5 keywords. In the current state, there are too many.
R: As suggested, the number of keywords has now been reduced to 5 as follows:

Keywords: SARS-CoV-2; transmembrane spike glycoprotein; severe pneumonia; natural compounds; antiviral activity 
2-    Figure 1 has a low resolution. Increase the resolution of the figure or replace it with a higher-quality image.
R: The quality of this figure has been improved. 
3-    The introduction is well-written, but there must be explained in some parts that the following references can be used: “Green synthesis of Mg0. 99 Zn0. 01O nanoparticles for the fabrication of κ-Carrageenan/NaCMC hydrogel in order to deliver catechin. Polymers, 12(4), 861”. And also “Effect of zinc content on structural, functional, morphological, and thermal properties of kappa-carrageenan/NaCMC nanocomposites. Polymer Testing, 93, 106922.”
R: Two references have been added to the introduction section as suggested (References 18 and 19). 
4-    Table 1 needs relevant references for each line.
R: All reference (38 to 50) have been added to table 1. 
5-    Clarify the novelty of the research.
R: We have improved the novelty of the research by adding the AutoDock Vina protocol for molecular docking stating the use of Nsp15 from SARS-CoV-2 (PDB: 6VWW) to the study. 
6-    Line 99, what is UCSF? Write its complete form for the first time using. Explain about UCSF Chimera1.12 program, and in the Materials section, there is no need to use references or links.
R: We have added this sentence ”UCSF Chimera1.12 program (a program for the interactive analysis and visualization of molecular structures including trajectories, sequence alignments, and density maps) was employed.”
7-    The resolution of all figures is too low. Draw the structures again and present them in high resolution.
R: The quality of all figures has been improved. 
9-    There are some old references that can be replaced with newly published papers.
R: We have replaced old references with new references (2020 to 2023). 

Reviewer 2 Report

In this article Mohammadabadi and coworkers conduct a docking study of the polyphenol epigallocatechin gallate and quercetin inside the spike (S) glycoprotein of SARS-CoV-Z.

Given that this is a very relevant topic nowadays, this work could be of interest as a starting point for a subsequent molecular dynamics study. Only for this reason we consider that this article could be published in Biomolecules with major revisions.

Line 210

The authors state:

"Based on the optimized geometry obtained by GaussView 5.0.8 software ..."

GaussView is only the graphical interfase for Gaussian. The optimization has been conducted with the Gaussian software.

Line 222

The authors state:

"..., higher density of electrons has been shown in red color compared to lower density of electrons ..."

The conclusions derived from the higher/lower electronic density are not discussed in depth.

Line 250

The authors state:

"In overall, this study showed that quercetin metabolite may be considered to be formulated as micro and nano antiviral drugs with doses of 500-1000 mg/day (or 50 and 75 mg/kg) in divided doses against SARS-CoV-2 infections."

It is not clear how this docking study can provide information about the quercetin dose against SARS-CoV-2 infections.

The quality of the figures should be improved.

Author Response

Dear Reviewer;
We thank you for your valuable suggestions and comments. Please find our responses to your comments / suggestions below (changes are highlighted in the manuscript in dark Green colour):

Line 210
The authors state:
"Based on the optimized geometry obtained by GaussView 5.0.8 software ..." 
GaussView is only the graphical interfase for Gaussian. The optimization has been conducted with the Gaussian software.
R: We have corrected the sentence as “Gaussian 5.0.8 software was employed to optimize geometry and to determine the electric field potential, lowest unoccupied molecular orbital (LUMO), and ……………." 

Line 222
The authors state:
"..., higher density of electrons has been shown in red color compared to lower density of electrons ..." 
The conclusions derived from the higher/lower electronic density are not discussed in depth.
 R: The conclusion section was improved as indicated by dark green highlight (for the comments of this Reviewer) and yellow (for the corrections done for Reviewer 3 comments). 

Line 250
The authors state:
"In overall, this study showed that quercetin metabolite may be considered to be formulated as micro and nano antiviral drugs with doses of 500-1000 mg/day (or 50 and 75 mg/kg) in divided doses against SARS-CoV-2 infections."
It is not clear how this docking study can provide information about the quercetin dose against SARS-CoV-2 infections.
R: In the introduction section, we have only presented the report about effective dose of quercetin, which by nano-formulation may be improved, not information about relationship between docking results with dose of quercetin. For this claim, in vivo study is needed. 

The quality of the figures should be improved.
R: We have improved the quality of all figures as suggested. 

Reviewer 3 Report

Although the article is interesting to read, some comments for the attention of the reviewers are listed below:

1. please improve the quality of the images for readability and clarity, especially in figures 1, 3, and 4 it is very hard to read any of the labels

2. Please correct the chemical formula of the compounds that appeared in the manuscript

3.  Would it be possible to include the IC50 of some of the reported inhibitors of EGCG  

4. Please include a section on the protocol for molecular docking stating the use of Nsp15 from SARS-CoV-2 (PDB: 6VWW) and AutoDock Vina. 

5. in line 136 the binding energy of this molecular docking was calculated as -7.287 kcal/mol for EGCG. However, many reports in the literature including ( PMID: 34015930) report −6.9 kcal/mol. could you please comment on the difference in the calculated docking value 

6. The bound EGCG molecule has hydrophobic interactions with Lys290, Val292, Tyr343, and Leu346 at the Nsp15 active site and forms hydrogen bonds with His235, Gly248, His250, Lys290, Ser294, and Thr341 at the site. The carbonyl group of EGCG has a polar interaction with Lys290 this is not shown in the model presented in the manuscript 

7. Structural docking studies previously suggested that the spike protein, PLpro, and RNA-dependent RNA polymerase may be possible targets of EGCG. this is not shown here 

8. Although there are many similar reports in the literature I am not sure what is the novelty of the manuscript 

9. Authors need to elaborate on the limitations of the study and also on the difficulty of the study design

10. The authors should also discuss the need for further experiments to be able to reach clinching conclusions

11. Some key references are missing from the manuscript relevant to articles (PMID: 32770567; PMID: 16387402; PMID: 28677813; PMID: 34015930; PMID: 36145270)

Author Response

Dear Reviewer;
We would like to thank you for your instructive suggestions. Please find our responses to your comments / suggestions below (changes are highlighted in the manuscript in Yellow colour):

1. please improve the quality of the images for readability and clarity, especially in figures 1, 3, and 4 it is very hard to read any of the labels.
R: We have improved the quality of these figures (1, 3, and 4). 

2.    Please correct the chemical formula of the compounds that appeared in the manuscript
R: All chemical formula has been corrected as yellow highlight.

3.    Would it be possible to include the IC50 of some of the reported inhibitors of EGCG.
R: In a comparative study, gallocatechin gallate, EGCG, quercetin, puerarin, and daidzein flavonoids exhibited 47, 73, 73, 381, and 351 µM of IC50 (50% inhibitory concentrations or half-maximal effective concentration), respectively for inhibition of SARS-CoV replication. Furthermore, docking scores of -14.1, -11.7, -10.2, -11.3, and -8.6 have been found for gallocatechin gallate, EGCG, quercetin, puerarin, and daidzein, respectively (Nguyen et al., 2012). In another study, EGCG had an IC50 value of 0.874 µM with the binding energy of −7.9 kcal/mol against 3CLpro SARS-CoV-2 (Du et al., 2021). 

Nguyen, T.T.H.; Woo, H.J.; Kang, H.K.; Nguyen, V.D.; Kim, Y.M.; Kim, D.W.;  Ahn, S-A.; Xia, Yongmei.; Kim, D. Flavonoid-mediated inhibition of SARS coronavirus 3C-like protease expressed in Pichia pastoris. Biotechnol. Lett. 2012; 34, 831-838. doi:10.1007/s10529-011-0845-8 
Du, A.; Zheng, R.; Disoma, C.; Li, S.; Chen, Z.; Li, S.; Liu, P.; Zhou, Y.; Shen, Y.; Liu, S.; et al. Epigallocatechin-3-gallate, an active ingredient of Traditional Chinese Medicines, inhibits the 3CLpro activity of SARS-CoV-2. Int. J. Biol. Macromol. 2021, 176, 1–12. 

4.    Please include a section on the protocol for molecular docking stating the use of Nsp15 from SARS-CoV-2 (PDB: 6VWW) and AutoDock Vina. 
R: Based on your comment, the protocol for molecular docking stating the use of Nsp15 from SARS-CoV-2 (PDB: 6VWW) and AutoDock Vina have been added to the manuscript. 

5.    in line 136 the binding energy of this molecular docking was calculated as -7.287 kcal/mol for EGCG. However, many reports in the literature including ( PMID: 34015930) report −6.9 kcal/mol. could you please comment on the difference in the calculated docking value 
R: This deference is not significant between different docking servers or software. In this regard, we have tried to compare these servers with AutoDock Vina due to getting comprehensive results. 

6.    The bound EGCG molecule has hydrophobic interactions with Lys290, Val292, Tyr343, and Leu346 at the Nsp15 active site and forms hydrogen bonds with His235, Gly248, His250, Lys290, Ser294, and Thr341 at the site. The carbonyl group of EGCG has a polar interaction with Lys290 this is not shown in the model presented in the manuscript 
R: This type of interaction has been indicated for EGCG towards 6VWW. We have added this section to the manuscript as yellow highlight: As illustrated in Figure 4a and 4b, LYS71 (H-bond), LYS90, GLY165, VAL166 (H-bond), THR167 (H-bond), ARG199, ASN200, GLU203, ASP268, ILE270, PRO271, MET272 (H-bond), ASP273, SER274, LYS277 (H-bond), and TYR279 (H-bond) amino acids of 6VWW contributed in the interaction with EGCG.

7.    Structural docking studies previously suggested that the spike protein, PLpro, and RNA-dependent RNA polymerase may be possible targets of EGCG. this is not shown here 
R: Based on this comment “Please include a section on the protocol for molecular docking stating the use of Nsp15 from SARS-CoV-2 (PDB: 6VWW) and AutoDock Vina” we have added the protocol for molecular docking stating the use of Nsp15 from SARS-CoV-2 (PDB: 6VWW) and AutoDock Vina to the manuscript.

8.    Although there are many similar reports in the literature I am not sure what is the novelty of the manuscript 
R: For first time, we have compared three docking server CB-Dock, EDock, and DockThor with VinaDock Vina for two main flavonoids compounds against Nsp15 and the spike protein of SARS-CoV-2.

9.    Authors need to elaborate on the limitations of the study and also on the difficulty of the study design
R: In the conclusion section, we have addressed the limitations of the study and also on the difficulty of the study design.

10.    The authors should also discuss the need for further experiments to be able to reach clinching conclusions
R: In the conclusion section, we have addressed all suggested points by 2 Reviewers. 

11.    Some key references are missing from the manuscript relevant to articles (PMID: 32770567; PMID: 16387402; PMID: 28677813; PMID: 34015930; PMID: 36145270)
R: We have added these references:
* Hashemzaei, M.; Delarami Far, A.; Yari, A.; Heravi, RE.; Tabrizian, K.; Taghdisi, SM.; Sadegh, SE.; Tsarouhas, K.; Kouretas, D.; Tzanakakis, G.; Nikitovic, D.; Anisimov, NY.; Spandidos, DA.; Tsatsakis, AM.;  Rezaee, R. Anticancer and apoptosis‑inducing effects of quercetin in vitro and in vivo. Oncol Rep. 2017;38(2):819-28. doi:10.3892/or.2017.5766
* Gasmi, A.; Mujawdiya, PK.; Lysiuk, R.; Shanaida, M.; Peana, M.; Gasmi Benahmed, A.; Beley, N.; Kovalska, N.;  Bjørklund, G. Quercetin in the Prevention and Treatment of Coronavirus Infections: A Focus on SARS-CoV-2. Pharmaceuticals (Basel). 2022;15(9). doi:10.3390/ph15091049
* Maiti, S.;  Banerjee, A. Epigallocatechin gallate and theaflavin gallate interaction in SARS-CoV-2 spike-protein central channel with reference to the hydroxychloroquine interaction: Bioinformatics and molecular docking study. Drug Dev Res. 2021;82(1):86-96. doi:10.1002/ddr.21730
* Isbrucker, RA.; Edwards, JA.; Wolz, E.; Davidovich, A.;  Bausch, J. Safety studies on epigallocatechin gallate (EGCG) preparations. Part 2: dermal, acute and short-term toxicity studies. Food Chem Toxicol. 2006;44(5):636-50. doi:10.1016/j.fct.2005.11.003
* Hong, S.; Seo, SH.; Woo, SJ.; Kwon, Y.; Song, M.;  Ha, NC. Epigallocatechin Gallate Inhibits the Uridylate-Specific Endoribonuclease Nsp15 and Efficiently Neutralizes the SARS-CoV-2 Strain. J Agric Food Chem. 2021;69(21):5948-54. doi:10.1021/acs.jafc.1c02050

Round 2

Reviewer 1 Report

The manuscript is improved and can be published in its current state.

Reviewer 2 Report

Now, the manuscript is OK for me.

Reviewer 3 Report

No Further comments to the authors